# Predictive modeling of apple slice drying: Integrating temperature, thickness, and shrinkage dynamics

**Mehdi Moradi** [ID]**\*, Reza Raeesi, Sadegh Rashidi, Mahdi Keramat-Jahromi** [ID]

Biosystems Engineering Department, School of Agriculture, Shiraz University, Shiraz, Iran

\* moradih@shirazu.ac.ir, mehdimoradi.ir@gmail.com

## Abstract

This study investigates the effects of drying temperature and slice thickness on the drying kinetics, shrinkage, moisture diffusivity, and color of Golden Delicious apple slices. Drying experiments were conducted in triplicate at 50°C, 60°C, and 70°C with slice thicknesses of 2, 4, and 6 mm using a controlled cabinet dryer. Statistical analysis using ANOVA confirmed that both factors significantly affected drying behavior and quality attributes. Drying time decreased with higher temperatures and thinner slices, with slice thickness exerting the stronger influence. Shrinkage decreased and effective moisture diffusivity increased under these conditions. Multivariate regression models accurately predicted shrinkage and moisture diffusivity ($R^2 > 0.97$), while Finite Element Modeling (FEM) closely matched experimental moisture transfer. Color evaluation showed that the combination of 70°C and a 4-mm slice thickness produced the lowest total color change (ΔE). Although high temperatures can accelerate browning, in this case the shorter drying time at 70°C limited discoloration, making this specific condition optimal for visual quality. These findings provide practical guidance for designing energy-efficient drying processes that maintain product quality, while future work should explore anisotropic shrinkage modeling, energy use, and hybrid drying technologies to enhance process performance.

## Introduction

Drying is one of the most traditional and widely used methods for food preservation. It significantly reduces moisture content to inhibit bacterial growth and prolong shelf life [1]. Over time, drying techniques have evolved from traditional sun drying to advanced methods such as infrared drying, hot air drying, and hybrid systems. These advancements aim to enhance efficiency, improve product quality, and reduce energy consumption [2]. Apples are among the most consumed fruits globally, valued for their nutritional benefits. They are rich in fiber, vitamins, minerals, and polyphenols— antioxidants that may help prevent various health conditions [3]. Efficient drying of

**Data availability statement:** All relevant data are within the paper and its Supporting information files.

**Funding:** The author(s) received no specific funding for this work.

**Competing interests:** The authors have declared that no competing interests exist.

apple slices is crucial to maintain their quality, including texture, color, and nutritional content. Research indicates that drying parameters, particularly slice thickness and temperature, significantly influence both the drying process and the quality of dried apple slices [4]. Higher drying temperatures accelerate moisture removal and reduce drying time but may cause undesirable changes in color, texture, and nutrient retention. Similarly, slice thickness significantly influences drying kinetics: thicker slices require longer drying times and can lead to uneven moisture distribution, quality degradation, or case hardening [5]. Mathematical modeling serves as an essential tool for understanding and optimizing food drying processes [6]. Finite Element Modeling (FEM) has been effectively employed to simulate heat and mass transfer during drying, providing insights into moisture diffusion and temperature distribution within food products. These models help predict drying behavior and refine process parameters to achieve target product quality and energy efficiency [7,8]. This study investigates the effects of slice thickness and drying temperature on the drying kinetics, moisture diffusivity, shrinkage, and color of apple slices. A key innovation of this work lies in the integration of experimental measurements with predictive modeling, combining multivariate regression and Finite Element Modeling (FEM) to simultaneously quantify multiple aspects of the drying process. Predictive models were developed to express shrinkage and moisture diffusivity explicitly as functions of slice thickness and temperature, providing a reliable, quantitative tool for optimizing drying parameters and improving both efficiency and product quality. Unlike previous studies, which often focus on individual factors or qualitative trends, this integrated approach enables accurate prediction of physical and quality-related changes under varying operational conditions, offering actionable insights for designing energy-efficient drying systems tailored to specific product characteristics.

## Materials and methods

### Drying system

Drying experiments were performed using a pilot-scale cabinet dryer equipped with an electric blower, heater, automatic weighing system, and chamber temperature control (Fig 1).

To ensure proper airflow, a centrifugal blower (Guangdong Shunde Electric Motor Co., Ltd.) with adjustable rotational speed was used. Operating at 1400 rpm and 23 W, the blower was connected to a rectangular duct (25 × 10 cm²) at the outlet. Downstream of the blower, a 1500 W heater with dimensions of 198 × 35 × 38 mm was installed.

To ensure that airflow distribution inside the drying chamber was uniform, airflow velocity was measured prior to the experiments using a digital hot-wire anemometer (Testo 405i, accuracy ±0.03 m/s). Measurements were taken at nine positions across the tray area (three rows × three columns) at the height of the apple slices. The airflow showed a variation of less than ±5% around the mean value of 1 m/s, confirming that the drying chamber provided sufficiently uniform airflow for consistent heat and mass transfer conditions during experiments. A temperature control module regulated the chamber temperature within a defined

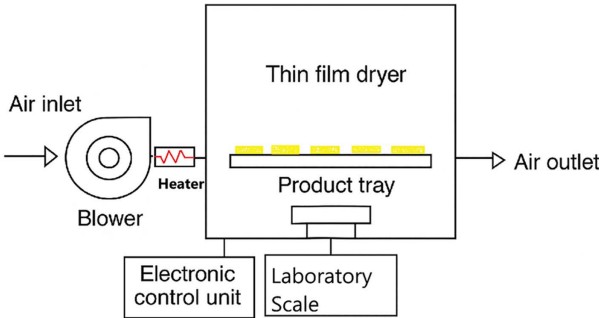

**Fig 1. Schematic diagram of the drying system used in this study.**

range. After setting the maximum and minimum temperature limits, the module was activated. The relay would switch on if the temperature fell below or equaled the minimum set point, and switch off if it exceeded the maximum set point. This circuit includes a relay capable of handling currents up to 7 amps, powered by a 12V supply. The temperature control board features a waterproof thermometer with an extended cable, allowing users to define specific temperatures for activating or deactivating the circuit. Two 220V, 120W inline AC dimmers were used to adjust the blower's rotational speed and the heater's voltage within the desired range. These dimmers were installed in the circuit paths of the blower and heater, enabling users to manually adjust their operating voltage to match the required range. A laboratory-grade scale (model GF3000, A&D Japan) with a precision of 0.001 g and a maximum capacity of 610 g was employed for the real-time weighing of samples. Data from the scale were directly transmitted to a computer via USB and saved at user-defined intervals to measure the moisture content of the drying sample. Relative humidity within the drying chamber was recorded using DHT22 or AM2302 temperature-humidity sensors installed at the chamber's inlet and outlet. Data from these sensors were displayed on a screen and transmitted to a computer via an Arduino MEGA 2560 microcontroller for recording and analysis. The data from the experiments were processed with IBM SPSS Statistics 27, and Duncan's test was used to evaluate the variance between the means. After setting up the system, the drying experiments were carried out in June 2024 at the Department of Biosystems Mechanical Engineering, Shiraz University. Fresh yellow apples (Golden Delicious variety) were procured daily from local market. The Golden Delicious apple was chosen due to its global commercial relevance, uniform morphology, and stable chemical composition, which minimize variability and make it a suitable model cultivar for evaluating drying performance and quality changes. The initial moisture content of the apples was measured by slicing the samples, weighing them with a precision scale (A&D, accuracy: 0.001 g), and placing them in a vacuum oven at 105°C for 24 hours. For each drying experiment, 25 g of freshly sliced apples with specified thicknesses and a consistent diameter of 43 mm (±2 mm) were placed in the dryer. The study was conducted at three different drying temperatures (50°C, 60°C, and 70°C) and three slice thicknesses (2 mm, 4 mm, and 6 mm) with an airflow speed of 1 m/s measured in the drying chamber. The experiments followed a factorial design with a completely randomized layout and were conducted in triplicate.

Shrinkage was calculated by measuring the diameter and thickness of apple slices before and after drying using a digital caliper with 0.01 mm accuracy. The product tray was positioned on a digital scale with 0.001 g precision, and the product weight was automatically recorded every 10 minutes. The moisture content and shrinkage were determined using Equations 1 and 2 [9].

$$MC_1 = \frac{(MC_0 \times W_0) - W_0 + W_1}{W_1}$$

(1)

$$Sh = \left[1 - \frac{V}{V_0}\right] \times 100$$

(2)

Where;

MC$_1$: Instantaneous moisture content of the sample, MC$_0$: Initial moisture content of the sample, W$_0$: Initial weight of the sample (kg), W$_1$: Instantaneous weight of the sample (kg), Sh: Shrinkage percentage, V: Volume of the dried product (m$^3$), V$_0$: The initial volume of the same sample prior to drying (m$^3$).

**Color change analysis.** To assess the color changes, apple slices were photographed before and after the drying process using a custom-built imaging box. A Xiaomi 5G camera equipped with a 64-megapixel wide-angle lens was used to capture the images. The images were analyzed in MATLAB R2018b, where Lab color space values were extracted for each sample. The total color difference (ΔE) was then determined using these values, as described in Equation 3 [10].

$$\Delta E = \sqrt{(\Delta L)^2 + (\Delta a)^2 + (\Delta b)^2}$$

(3)

Where ΔL, Δa, and Δb represent the differences in the color parameters before and after drying.

**Theoretical modeling.** The transient mass-transfer equation in a two-dimensional plane is a parabolic partial differential equation (Equation 4). Analytical solutions to this equation are only possible when the effective moisture diffusivity is assumed to be constant, which is a common simplification in classical drying theory. However, in real food tissues such as apples, the diffusivity changes during drying due to structural collapse, shrinkage, and evolving moisture gradients; therefore, a constant-diffusivity assumption is not appropriate.

For this reason, the present study employs a variable effective diffusivity, and the equation is solved numerically using the Finite Element Method (FEM). The FEM formulation was implemented with the Galerkin Weighted Residual approach, in which the weighted residual integrals over each element are set to zero. Applying Green's theorem and performing the necessary simplifications yields the numerical solution for the spatial and temporal moisture distribution. Full methodological details are provided in [11].

$$\frac{\partial M}{\partial t} = D_{eff}\left[\frac{\partial^2 M}{\partial x^2} + \frac{\partial^2 M}{\partial y^2}\right]$$

(4)

Where: $D_{eff}$: Effective moisture diffusivity (m$^2$/s), M: Moisture content (dry basis), x,y: Dimensions of apple slices (mm).

Convective boundary conditions were applied to the mass transfer equation Equations 5 and 6 [12,13].

$$-D_{eff}\left(\frac{\partial M}{\partial x}\right)_{x=x_0} = h_D\left(M_{surf} - M_e\right)$$

(5)

$$-D_{eff}\left(\frac{\partial M}{\partial y}\right)_{y=y_0} = h_D\left(M_{surf} - M_e\right)$$

(6)

Where: $h_D$: Coefficient of convective mass transfer (m/s), $M_{surf}$: Surface moisture content, $M_e$: Equilibrium moisture content.

In a study on modeling heat and mass transfer during the convective drying of fruits, a convective boundary condition was applied [14].

Similarly, in another study on thin-layer drying of cumin seeds, the convective boundary condition was used to simulate the drying process [13].

Equilibrium moisture content was obtained using psychrometric charts based on relative humidity and temperature. The convective mass transfer coefficient ($h_D$) was calculated using correlations Equations 7–10, given that the Reynolds number was below $3 \times 10^5$ [15]:

$$Sh = 0.664Re^{0.5}Sc^{0.33} \tag{7}$$

$$Re = \frac{\rho V d}{\mu} \tag{8}$$

$$Sh = \frac{h_D d}{D} \tag{9}$$

$$Sc = \frac{\nu}{D} \tag{10}$$

These correlations were selected because the airflow in the drying chamber produced Reynolds numbers below $3 \times 10^5$, which fall within the laminar–transitional regime where Sherwood–Reynolds–Schmidt correlations are applicable. Apple slices are thin, smooth, and approximately flat, making their geometry compatible with the flat-plate assumptions underlying these correlations. Such correlations have been successfully applied in previous drying studies of fruits and vegetables with similar geometries and airflow conditions, supporting their suitability for estimating the convective mass-transfer coefficient (hD) in this study. For example, Tuly et al. [16] calculated the mass-transfer coefficient using the Sherwood number (Sh) and Schmidt number (Sc) as given in Equations (7–10). Additionally, Kumar et al. [17] developed a multiphysics drying model in which mass transfer was solved numerically with variable properties, employing Sherwood–Reynolds–Schmidt type correlations.

Moisture diffusivity ($D_{eff}$) of apple slices was calculated using Equation 11 according to a previous described study [18].

$$\ln MR = \ln\left(\frac{8}{\pi^2}\right) - \left(\frac{\pi^2 D_{eff} t}{4L^2}\right) \tag{11}$$

Where: $MR = M/M0$, t: Drying time (s), L: Sample thickness (m).

By graphing ln MR versus t, the slope (k) can be determined, which is then used to calculate $D_{eff}$ (Equation 12): [18].

$$k = \frac{-\pi^2 D_{eff}}{4L^2} \tag{12}$$

The Fickian diffusion equation was subsequently solved using MATLAB 2024a software.

The temperature dependence of moisture diffusion was modeled using an Arrhenius-type equation (Equation 13): [18].

$$\ln(D_{eff}) = \ln(D_0) - \frac{E_a}{R} \times \frac{1}{T_{abs}} \tag{13}$$

Where, $E_a$ is activation energy (kJ/mol), $D_0$ is the pre-exponential factor (m²/s), $T_{abs}$ is absolute temperature (K), and R is the universal gas constant (8.314 J/mol·K).

**Measurement uncertainty and error propagation.** To evaluate the reliability of the experimental results and derived parameters, uncertainty analysis was performed using a standard error propagation method. When a

parameter Y depends on multiple measured variables $a_1, a_2, ..., a_n$, the combined uncertainty $\delta Y$ was estimated using the Equation 14: [19].

$$\delta Y = \sqrt{\left(\frac{\partial Y}{\partial a_1}.\delta a_1\right)^2 + \left(\frac{\partial Y}{\partial a_2}.\delta a_2\right)^2 + ... + \left(\frac{\partial Y}{\partial a_n}.\delta a_n\right)^2}$$

(14)

This method accounts for the sensitivity of the output to each input variable and propagates the individual measurement uncertainties accordingly. It was applied to key calculated parameters such as moisture content, shrinkage, and total color change (ΔE). For example, the uncertainty in Deff was estimated based on the errors in slice thickness and the slope obtained from drying curves, while ΔE uncertainty was calculated using the propagated errors in L*, a*, and b* values from image analysis.

The measurement uncertainties used in these calculations were derived from manufacturer specifications and validated through repeated trials. A summary of the instruments and their corresponding uncertainties is provided in Table 1.

## Results and discussions

### Drying kinetics

The effects of drying temperature (50, 60, and 70°C) and apple slice thickness (2, 4, and 6 mm) on time required to reduce moisture content from an initial average of 82.5±1% to 12±1% (dry basis) were investigated. The results of the variance analysis are shown in Table 2.

**Table 1. Uncertainties and Errors During the Experiment.**

| Instrument | Error |
|---|---|
| Digital Caliper (slice thickness) | ±0.01 mm |
| Electronic Scale (A&D GF3000) | ±0.001 g |
| Thermocouple (drying air temp) | ±0.5 °C or ±2.0% |
| Humidity Sensor (DHT22) | ±2.0% RH |
| Anemometer (air velocity) | ±3.0% |
| Drying time | ±10 s |
| Camera (color imaging) | ±1.5 ΔE (estimated) |
| Parameters | Uncertainty |
| Moisture Content | ±0.99% |
| Shrinkage | ±1.2% (combined) |
| Color Change (ΔE) | ±1.5 |

**Table 2. Anova results for the effect of thickness and temperature on drying time.**

| Variation source | Df | Sum square | Mean square | F | P |
|---|---|---|---|---|---|
| Corrected model | 8 | 3600 | 4575 | 183** | 0.0001 |
| Intercept | 1 | 357075 | 357075 | 14283** | 0.0001 |
| T | 2 | 8600 | 4300 | 172** | 0.0001 |
| H | 2 | 26600 | 13300 | 532** | 0.0001 |
| T×H | 4 | 1400 | 350 | 14** | 0.0001 |
| Error | 18 | 450 | 25 | | |
| Total | 27 | 394125 | | | |
| Corrected total | 26 | 37050 | | | |

**Analysis of drying time and kinetics.** The effects of temperature (p < 0.01), slice thickness (p < 0.01), and their interaction (p < 0.01) on drying time were statistically significant. Given that the F-value for slice thickness was higher than for temperature, slice thickness had a more pronounced impact on the drying time of apple slices than temperature. Evidence from previous drying studies supports the greater influence of slice thickness compared to temperature on drying time. For instance, in the drying of melon slices, increasing thickness from 3 mm to 5 mm led to increases of 38–87% in drying time across temperatures of 60, 70, 80, and 90°C, whereas raising the air temperature within a constant thickness reduced drying time by only 40–53% [20]. Similarly, in studies on other fruit matrices such as *G. erubescens*, thicker slices consistently required substantially longer drying periods due to the increased moisture diffusion path [21]. These findings collectively demonstrate that slice thickness exerts a stronger limiting effect on moisture removal than temperature, reinforcing our observation that thickness has a more pronounced impact on drying time. The interaction effect between temperature and thickness (F-value = 14) further confirmed a significant non-additive relationship. Specifically, the influence of slice thickness became less pronounced at higher drying temperatures. Previous studies align with these findings: One study reported a highly significant effect of temperature on drying kinetics in green bell pepper slices using a spouted bed dryer, where increased temperature substantially reduced drying time [9]. Similarly, another study found that raising the drying temperature from 45°C to 65°C while decreasing apple slice thickness from 5 mm to 1.5 mm significantly shortened drying times [21]. Thus, optimizing drying requires simultaneous consideration of both factors. Higher temperatures are particularly effective for thicker slices, as they mitigate the impact of thickness on drying time. A mean comparison of the effects of temperature, slice thickness, and their interaction on drying time was conducted, as presented in Fig 2. The analysis was based on Duncan's multiple range test, and the results indicate that all levels of the independent variables differ significantly from one another.

The drying kinetics of apple slices at temperatures of 50°C, 60°C, and 70°C respectively illustrated in the Figs 3–5.

Drying time decreased with increasing temperature and decreasing slice thickness, with thicker slices requiring substantially longer times than thinner ones. At 50°C, 60°C, and 70°C, the trends are consistent across all thicknesses, as illustrated in Fig 6. Higher temperatures effectively mitigate the impact of slice thickness on drying time, especially for thicker slices. Fig 6 illustrates the drying time under these conditions, showing that higher drying temperatures and thinner slices significantly reduced drying time. Accordingly, at 50°C, increasing thickness from 2 to 4 mm resulted in a 67% longer drying time, while increasing from 4 to 6 mm added another 26%. At 60°C, these increases were 57% and 27%, respectively. At 70°C, increasing thickness from 2 to 4 mm extended drying time by 60%, while the increase from 4 to 6 mm was 50%. These results demonstrate that slice thickness has a more pronounced effect on drying time at lower temperatures.

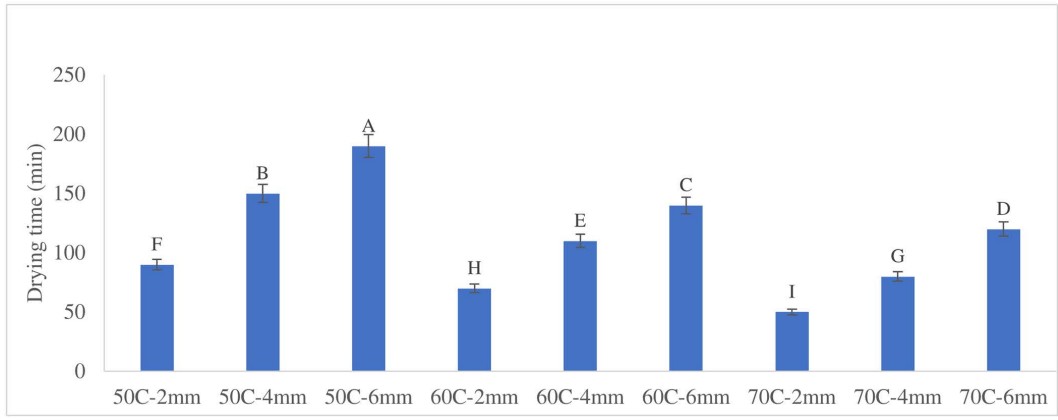

**Fig 2. Mean comparison of drying time under different drying treatments.** The letters show significant difference at P-value < 0.01.

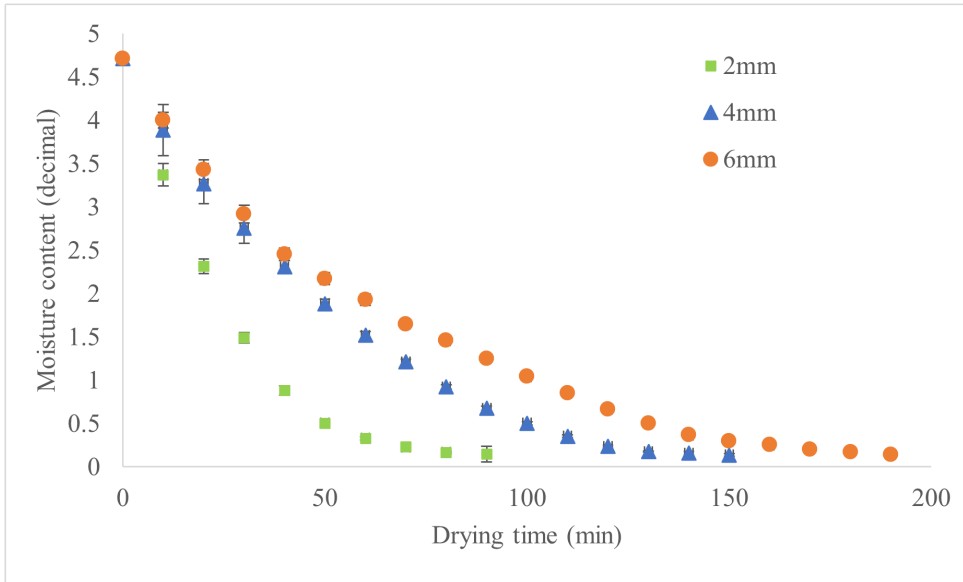

**Fig 3. Moisture Content of Drying Samples at 50°C for Various Slice Thicknesses.**

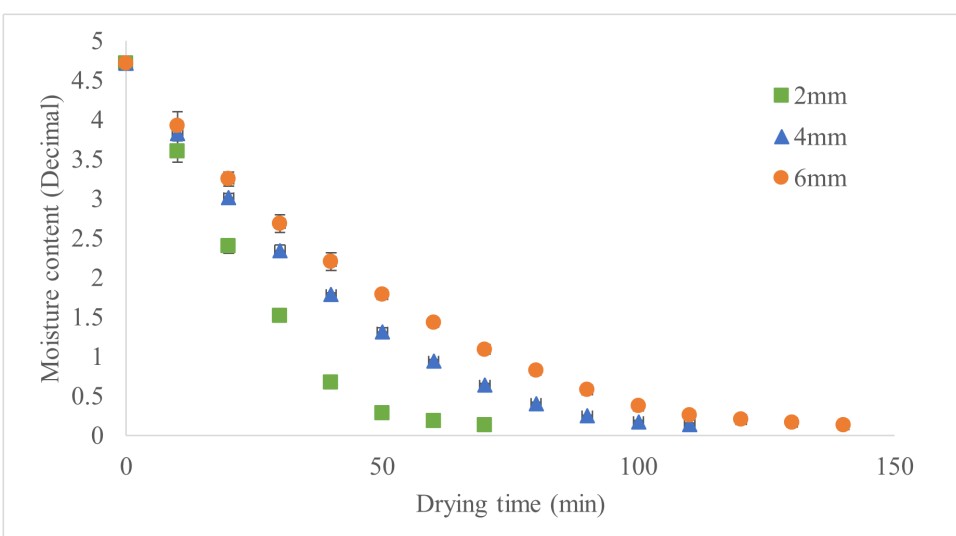

**Fig 4. Moisture Content of Drying Samples at 60°C for Various Slice Thicknesses.**

Hence, applying a slice thickness of 2 mm at a drying temperature of 70 °C resulted in the shortest drying time, reducing it by approximately 74% compared to the treatment with the longest drying time (6 mm at 50 °C). Another study reported that higher temperatures and thinner slices increase the drying rate and effective moisture diffusivity, resulting in more efficient drying processes [21].

The drying characteristics of apple slices show distinct patterns depending on slice thickness (2, 4, and 6 mm) and drying temperature (50°C, 60°C, and 70°C). At 50 °C, thicker slices (6 mm) showed slower moisture removal than thinner slices (2 mm). At 60°C, the drying process was faster, with shorter drying times and a reduced influence of slice thickness,

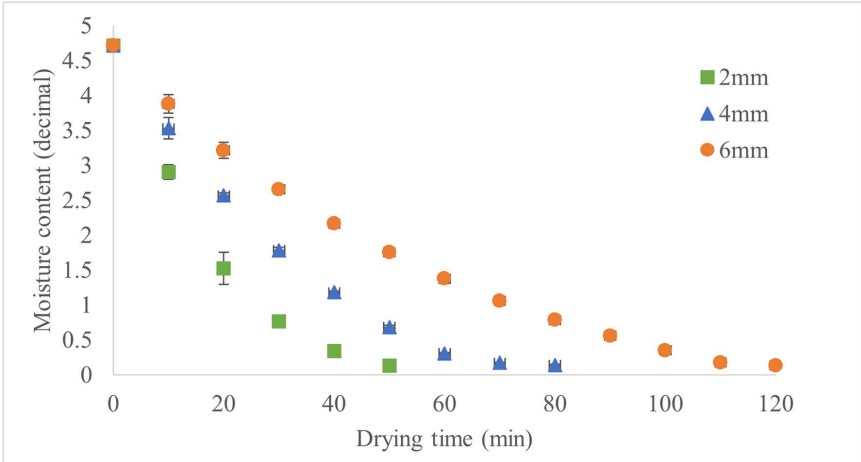

**Fig 5. Moisture Content of Drying Samples at 70°C for Various Slice Thicknesses.**

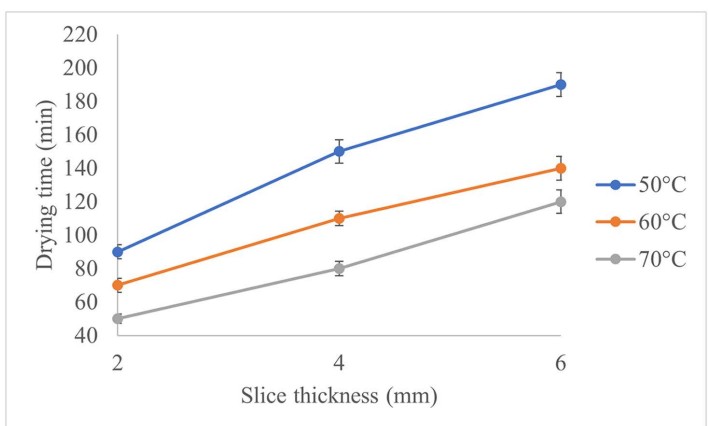

**Fig 6. Drying Times for Various Experiments.**

likely due to improved moisture diffusion. At 70°C, drying time was the shortest, with rapid moisture reduction across all slice thicknesses, and the differences in drying time due to thickness were minimized. In another study, vacuum drying experiments showed that increasing the temperature from 50 °C to 70 °C nearly halved the drying time for 5 mm thick apple slices, though the reduction was less significant for 7 mm slices [22].

**Shrinkage analysis.** Table 3 presents the analysis of variance (ANOVA) for the effects of slice thickness and temperature on the shrinkage of apple slices. The results indicate that both temperature and thickness significantly influenced shrinkage ($p < 0.01$(, with the effect of thickness being more substantial than that of temperature, as evidenced by the higher F-value for thickness.

Mean comparison, performed using Duncan's test, showed that all groups of the independent factors differed significantly from one another Fig 7.

Fig 8 illustrates the shrinkage values versus various drying temperatures. The results indicate that the shrinkage of apple slices decreases with increasing drying temperature and decreasing slice thickness. This reduction is likely due to shorter drying times at higher temperatures and thinner slices, which reduce the extent of shrinkage.

**Table 3. Anova results for the effect of temperature and thickness on shrinkage.**

| Variation source | Df | Sum square | Mean square | F | P |
|---|---|---|---|---|---|
| Corrected model | 8 | 0.248 | 0.031 | 5.648** | 0.0001 |
| Intercept | 1 | 1.775 | 325.002 | 325.002** | 0.0001 |
| T | 2 | 0.036 | 3.281 | 3.281** | 0.0001 |
| H | 2 | 0.089 | 8.191 | 8.191** | 0.0001 |
| T×H | 4 | 0.123 | 5.632 | 5.632** | 0.0001 |
| Error | 18 | 0.098 | | | |
| Total | 27 | 2.121 | | | |
| Corrected total | 26 | 0.347 | | | |

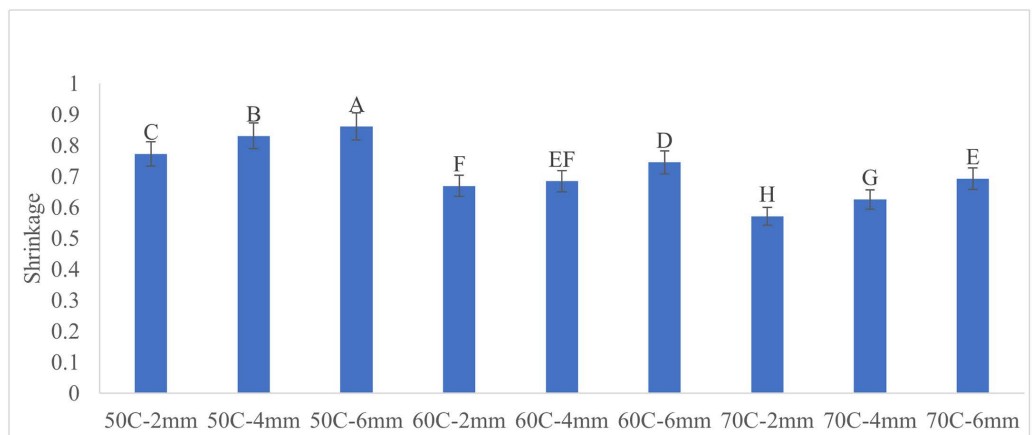

**Fig 7. Mean comparison of shrinkage under different drying treatments.** The letters show significant difference at P-value < 0.01.

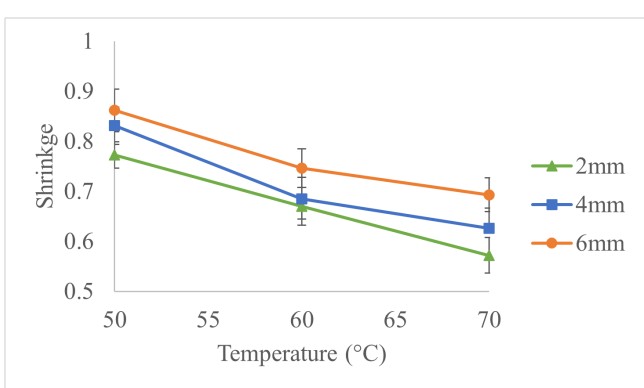

**Fig 8. Shrinkage Variations with Slice Thickness and Drying Temperature.**

The highest shrinkage value of 0.862 was observed for 6 mm slices dried at 50°C, while the lowest value of 0.572 was recorded for 2 mm slices dried at 70°C.

A separate study found that increasing microwave power levels reduced the shrinkage of apple slices during drying [23]. Similarly, another study examining the effect of slice thickness and temperature on apple slice shrinkage reported that rising the temperature from 60°C to 80°C, along with reducing slice thickness, led to decreased shrinkage [24].

Using the experimental data, a multivariate regression model was developed to predict shrinkage Equation 15:

$$S = 1.197 - 0.00958T + 0.0238H \qquad (15)$$

Where: S: Shrinkage, T: Drying temperature (°C), H: Slice thickness (mm).

The model, with a determination coefficient ($R^2$) of 0.97 and a standard error of 0.0198, effectively predicts shrinkage as a function of slice thickness and drying temperature. It is important to note that this model assumes a linear relationship; shrinkage may exhibit non-linear behavior at wider ranges of temperature or slice thickness due to phenomena such as structural collapse or moisture-dependent diffusivity. A comparison between experimental shrinkage values and model predictions is presented in Fig 9. Accordingly, increasing the temperature and decreasing the slice thickness effectively reduces shrinkage. Notably, the coefficient for slice thickness is approximately 2.5 times greater in magnitude than that for temperature, indicating that slice thickness exerts a more substantial influence on shrinkage. To minimize shrinkage during the dehydration of apple slices, it is recommended to use higher drying temperatures in combination with thinner slice thicknesses. A study has shown that increasing the drying temperature accelerates moisture removal, reducing the time available for structural changes that cause shrinkage. Additionally, thinner slices allow for faster moisture loss, which further contributes to reduced shrinkage [25].

**Moisture diffusivity.** Moisture diffusivity considering shrinkage during drying time, was calculated for various drying experiments. The mean moisture diffusivity values ranged between $1.08 \times 10^{-10}$ and $9.047 \times 10^{-10}$ m² s⁻¹, aligning with previously reported ranges for apples. The effective moisture diffusion coefficient during the drying of fruits varies depending on factors such as fruit type, drying temperature, slice thickness, and air velocity. Generally, these coefficients range from approximately $10^{-12}$ to $10^{-6}$ m² s⁻¹, with most values concentrated between $10^{-11}$ and $10^{-8}$ m² s⁻¹ [26]. Fig 10 depicts the average moisture diffusivity for the experiments. The highest moisture diffusivity was observed at 70°C, consistent with prior research showing that moisture diffusivity increases with temperature [27–29]. Additionally, as slice thickness increased, moisture diffusivity rose, potentially due to larger moisture concentration gradients in thicker slices.

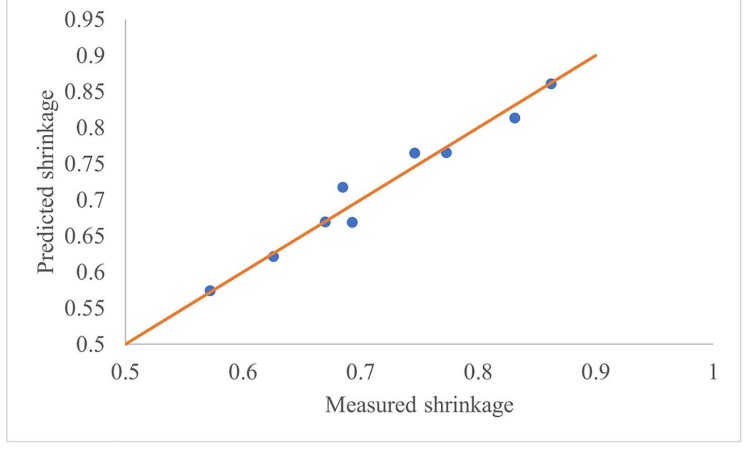

**Fig 9. Comparison of Experimental and Predicted Shrinkage Values.**

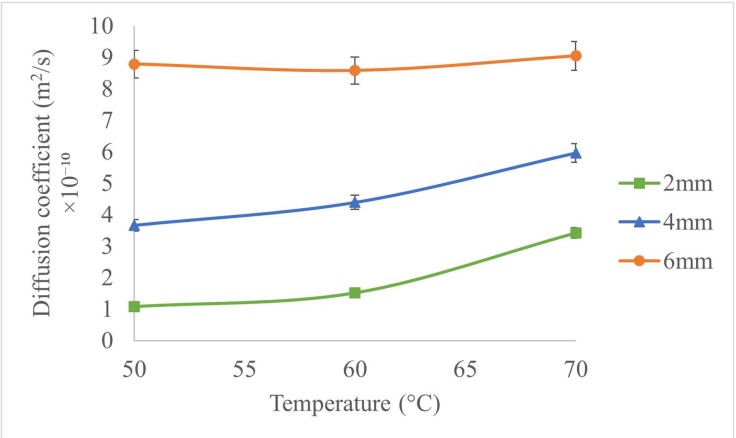

**Fig 10. Moisture Diffusivity of Apple Slices at Different Temperatures and Thicknesses.**

Although it may seem counterintuitive, effective moisture diffusivity increased with slice thickness due to larger internal moisture gradients in thicker slices, which enhance the driving force for moisture movement. This trend reflects average diffusivity over the drying period rather than local instantaneous values. In a study Royen et al. [5], moisture diffusivity in apple slices increased with slice thickness, ranging from 4 to 12 mm. Similarly, Limpaiboon [30]. reported that increasing slice thickness and temperature enhanced the moisture diffusivity of pumpkin slices.

A multivariate regression model was developed to express moisture diffusivity as a function of slice thickness and temperature Equation 16:

$$D_{eff} = 4.96 \times 10^{-13}T^2 - 5.13 \times 10^{-11}T + 1.82 \times 10^{-11}H^2 + 2.36 \times 10^{-11}H + 1.345 \times 10^{-9} \quad (16)$$

Where: T: Drying temperature (°C), H: Slice thickness (mm).

The model, with an $R^2$ of 0.98 and a standard error of $4.6 \times 10^{-11}$, accurately predicts moisture diffusivity within the experimental conditions. A comparison of the modeled and experimental moisture diffusivity values is shown in Fig 11. The model can be analyzed as follows:

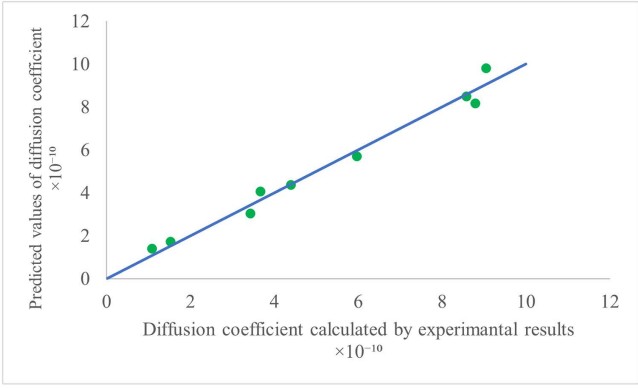

**Fig 11. Comparison of Predicted and Experimental Moisture Diffusivity Values.**

The model can be interpreted as follows:

1. The temperature terms (T² and T) together show that $D_{eff}$ changes in a non-linear way with temperature, increasing more at higher temperatures.

2. The slice thickness terms (H² and H) show that thicker slices have higher Deff, with the effect becoming stronger as thickness increases.

3. The coefficients for H (both linear and quadratic) are larger than those for T, indicating that slice thickness has a stronger impact on $D_{eff}$ than temperature. This aligns with the physical expectation that thicker slices develop larger internal moisture gradients, which enhance diffusivity.

**Activation energy.** The Arrhenius equation (Equation 13) was used to determine the activation energy for moisture diffusion across different slice thicknesses. By plotting $\ln(D_{eff})$ against $(1/T_{abs})$ and analyzing the slope, activation energies of 56.55, 35.23, and 20.75 kJ/mol were obtained for 2, 4, and 6 mm slices, respectively. The higher activation energy for thinner slices indicates that moisture diffusion in these slices is more sensitive to temperature changes than in thicker slices. The substantial differences in activation energy between thicknesses are primarily due to variations in structural resistance to moisture movement. Thinner slices have a higher surface-to-volume ratio and lose moisture rapidly near the surface, creating stronger resistance to continued internal moisture transport. Sustaining diffusion under these conditions requires more energy, resulting in higher activation energy values. In contrast, thicker slices exhibit slower, more uniform internal drying with less structural resistance, making their diffusivity less sensitive to temperature and thus lowering activation energy. Similar trends have been observed in other studies; for example, the activation energy for carrot pomace was 27.64 kJ/mol for 5 mm slices and 17.92 kJ/mol for 10 mm slices [31].

**Color analysis.** The ANOVA results (Table 4) indicate that drying temperature (T), slice thickness (H), and their interaction (T × H) all had significant effects on the total color difference (ΔE) of apple slices (p < 0.0001). The F-values suggest that temperature had the strongest individual effect on ΔE, followed by slice thickness, while the significant interaction term demonstrates that the effect of temperature on color change depended on slice thickness. Overall, the corrected model was highly significant, confirming that the variation in ΔE was largely explained by the combined drying factors.

Accordingly, the drying conditions had a significant impact on the color parameters (L*, a*, b*) of apple slices. In general, L* after drying across all treatments, with post-drying values ranging from 93 to 96, indicating a reduction in browning and an overall lighter appearance. This effect was particularly pronounced at higher temperatures (60°C and 70°C) and thinner slices (2 mm), which allowed for faster moisture removal and potentially minimized enzymatic browning reactions.

**Table 4. Anova results for the effect of temperature and thickness on color change.**

| Variation source | Df | Sum square | Mean square | F | P |
|---|---|---|---|---|---|
| Corrected model | 8 | 213.4 | 26.67 | 2667** | 0.0001 |
| Intercept | 1 | 1449.8 | 1449.8 | 145000** | 0.0001 |
| T | 2 | 51.13 | 25.6 | 2556** | 0.0001 |
| H | 2 | 23.14 | 11.6 | 1157** | 0.0001 |
| T × H | 4 | 139.12 | 34.8 | 3478** | 0.0001 |
| Error | 18 | 0.18 | 0.01 | | |
| Total | 27 | 1663.4 | | | |
| Corrected total | 26 | 213.6 | | | |

The average values of L*, a*, b* for all experiments at before and after the drying were shown in Table 5.

**Table 5. Average values of L*, a*, and b* color parameters of apple slices before and after drying under different drying treatments.**

| Drying Experiment | L* | a* | b* | ΔE |
|---|---|---|---|---|
| 50°C,2 mm-before drying | 86 | 0 | 11 | 8.12 |
| 50°C,2 mm-after drying | 94 | −1 | 12 | |
| 50°C,4 mm-before drying | 95 | −2 | 8 | 8.31 |
| 50°C,4 mm-after drying | 93 | −1 | 16 | |
| 50°C,6 mm-before drying | 97 | −2 | 7 | 11.36 |
| 50°C,6 mm-after drying | 95 | −4 | 18 | |
| 60°C,2 mm-before drying | 92 | 1 | 0 | 3.74 |
| 60°C,2 mm-after drying | 94 | 0 | 3 | |
| 60°C,4 mm-before drying | 96 | −1 | 5 | 6.08 |
| 60°C,4 mm-after drying | 96 | −2 | 11 | |
| 60°C,6 mm-before drying | 97 | 0 | 4 | 8.60 |
| 60°C,6 mm-after drying | 96 | −3 | 12 | |
| 70°C,2 mm-before drying | 84 | −1 | 4 | 11.45 |
| 70°C,2 mm-after drying | 95 | −2 | 7 | |
| 70°C,4 mm-before drying | 94 | −2 | 10 | 3.74 |
| 70°C,4 mm-after drying | 96 | −3 | 13 | |
| 70°C,6 mm-before drying | 99 | 0 | 3 | 4.58 |
| 70°C,6 mm-after drying | 95 | −1 | 5 | |

The a* values, which represent the red-green axis, slightly decreased (more negative), indicating a shift toward greener tones. This shift suggests suppression of browning reactions that typically push a* into the red region. The b* values, indicating yellowness, consistently increased after drying, ranging from 3 to 18, with more pronounced yellowing at lower drying temperatures (50°C) and thicker slices (6 mm), likely due to extended drying times that enhanced Maillard reactions or pigment concentration.

Accordingly, the drying conditions had a significant impact on the color parameters (L*, a*, b*) of apple slices. In general, L* after drying across all treatments, with post-drying values ranging from 93 to 96, indicating a reduction in browning and an overall lighter appearance. This effect was particularly pronounced at higher temperatures (60°C and 70°C) and thinner slices (2 mm), which allowed for faster moisture removal and potentially minimized enzymatic browning reactions. The a* values, which represent the red-green axis, slightly decreased (more negative), indicating a shift toward greener tones. This shift suggests suppression of browning reactions that typically push a* into the red region. The b* values, indicating yellowness, consistently increased after drying, ranging from 3 to 18, with more pronounced yellowing at lower drying temperatures (50°C) and thicker slices (6 mm), likely due to extended drying times that enhanced Maillard reactions or pigment concentration.

From a consumer acceptability perspective, lighter and less browned slices are typically preferred. According to Arendse & Jideani [32], dried apple slices pre-treated with citric acid and moringa leaf extract exhibited color values of L*=85.6, a*=1.5, and b*=17.9 at day 0, and these samples were rated highly for consumer acceptability of color. Most of our dried samples meet or exceed these values, suggesting good potential for consumer acceptance. However, samples such as 50°C-6 mm, which exhibited a higher b* value (18) and a* value of −4, may appear overly yellow, potentially affecting visual appeal. Overall, optimal drying conditions, particularly higher temperatures (60–70°C) combined with thin slices (2–4 mm), produced apple slices with favorable color characteristics that align well with consumer preferences.

Color changes under different drying conditions are illustrated in Fig 12 and can be explained by the interplay of enzymatic browning and Maillard reactions. At 50°C, ΔE increases with increasing sample thickness, with the highest color difference observed in the 6 mm slices. This trend is primarily due to enzymatic browning, which remains active longer at lower temperatures and in thicker slices where moisture is retained for extended periods. The prolonged drying time under these conditions allows for greater enzyme activity, resulting in more pronounced browning and thus higher ΔE values. This phenomenon is supported by Gao et al. [33], who reported that enzymatic browning is the primary cause of color changes during the initial stages of drying apple slices. At 60°C, thinner slices (2 mm) exhibit lower ΔE compared to thicker ones. This moderate temperature provides a balance where enzyme activity is partially deactivated, reducing enzymatic browning, while Maillard reactions are still limited due to insufficient thermal activation. As a result, overall color degradation is minimized, making this condition ideal for maintaining visual quality. In contrast, at 70°C, a different pattern emerges: ΔE is highest in the thinnest slices and lower in thicker ones. At this high temperature, enzymatic browning is quickly suppressed due to enzyme denaturation. However, the Maillard reaction becomes the dominant pathway, particularly in thinner slices where rapid drying concentrates sugars and amino acids near the surface, accelerating non-enzymatic browning. Thicker slices at this temperature retain internal moisture longer, which buffers the heat effect and slows down surface-level chemical reactions, resulting in lower ΔE values. Overall, these findings suggest that the dominant browning mechanism shifts from enzymatic to Maillard-based as drying temperature increases. For optimal color retention, drying at intermediate temperatures (around 60°C) with moderate slice thicknesses (4–6 mm) is recommended. In contrast, higher temperatures and thinner slices may be appropriate for applications where faster drying is prioritized over color preservation. These findings are corroborated by studies on other drying methods. For instance, in hot-air drying, significant increases in ΔE and browning index (BI) were observed, indicating substantial color changes due to non-enzymatic browning [34].

**Modeling.** To simulate the moisture concentration within the sample during drying, a triangular mesh was employed for discretization of the domain. Also, a mesh sensitivity analysis was conducted by simulating the drying process using four different mesh densities. The convergence of average moisture concentration was assessed by comparing results against the finest mesh. As shown in Fig 13 for a slice thickness of 2 mm the solution stabilized with mesh sizes finer than 0.0005 m, confirming mesh independence of the simulation. Error (E) was obtained from Equation 17:

$$E = \frac{|M_1 - M_0|}{M_0}$$

(17)

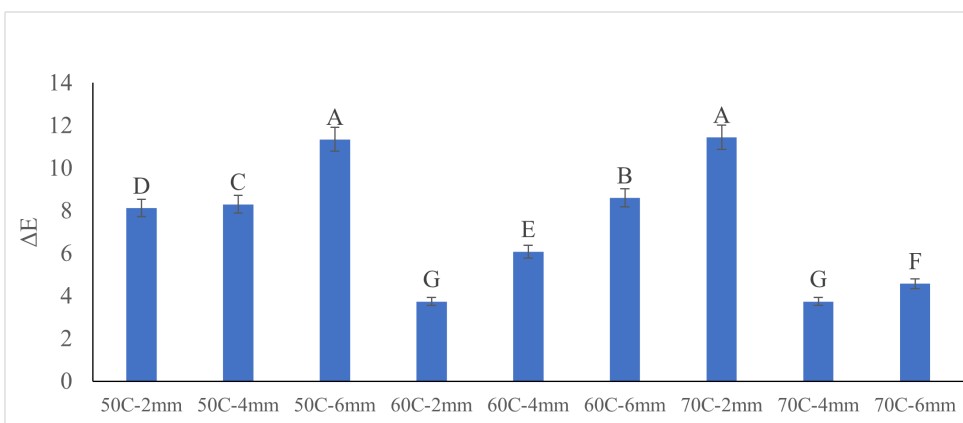

**Fig 12. Color Changes for Different Drying Conditions.** The letters show significant difference at P-value < 0.01.

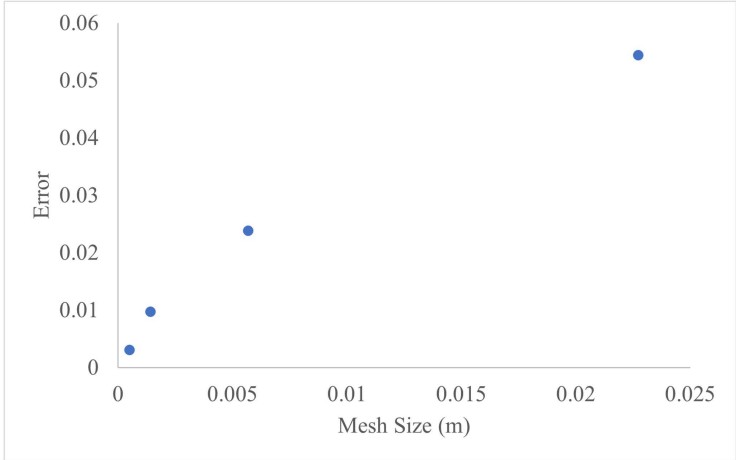

**Fig 13. Mesh size versus error of moisture content estimation.**

In which $M_1$ is average moisture content of sample estimated by each mesh size while $M_0$ is moisture content estimated by finest mesh size.

Comparisons between experimental and modeled average moisture content, by MATLAB software, are shown in Fig 14 for different drying conditions.

Table 6 presents the statistical metrics, such as Mean Absolute Error (MAE), Root Mean Square Error (RMSE), and $R^2$, for the comparison between experimental and modeled drying data.

The 2D drying model assumes isotropic shrinkage and uniform properties across the slice thickness. While this simplifies computation, it may not fully reflect the anisotropic shrinkage and moisture gradients present in real apple tissues, especially in thicker slices. These assumptions can lead to slight overestimations of drying uniformity and underestimations of internal moisture content. Despite this, the model provides a good balance between accuracy and efficiency for thin slices. Future work may benefit from incorporating anisotropic effects or extending the model to 3D.

## Statistical analysis

The theoretical model demonstrated a high correlation with experimental data, achieving $R^2$ values between 0.97 and 0.99. The model's accuracy was further confirmed by low Mean Absolute Error (MAE) values (0.063–0.186) and Root Mean Square Error (RMSE) values (0.092–0.275). Experimental slicing inaccuracies, estimation errors for moisture diffusivity, and boundary condition assumptions contributed to modeling errors. Nonetheless, the mean MAE of 0.12 (11% error) is acceptable for theoretical models. This aligns with findings in similar research where theoretical drying models yield high $R^2$ values (>0.95), confirming the validity of the drying kinetics model. For instance, Das and Prasad [8], investigated drying kinetics of bell peppers and achieved $R^2$ values between 0.94 and 0.98, with MAE values typically <0.15. Their findings also noted higher errors at lower temperatures due to prolonged drying times and more significant moisture gradients within the slices. From the FEM it can be concluded that thicker slices (6 mm) showed lower errors (e.g., MAE: 0.063 at 70°C) due to more uniform moisture diffusion. Thinner slices (2 mm) had higher errors (e.g., MAE: 0.186 at 60°C), likely due to greater susceptibility to boundary condition assumptions. Also, higher temperatures (70°C) improved accuracy as moisture diffusivity increased, leading to faster and more uniform drying. Lower temperatures (50°C) introduced larger errors due to slower drying rates and potential estimation inaccuracies for moisture diffusivity. In another study,

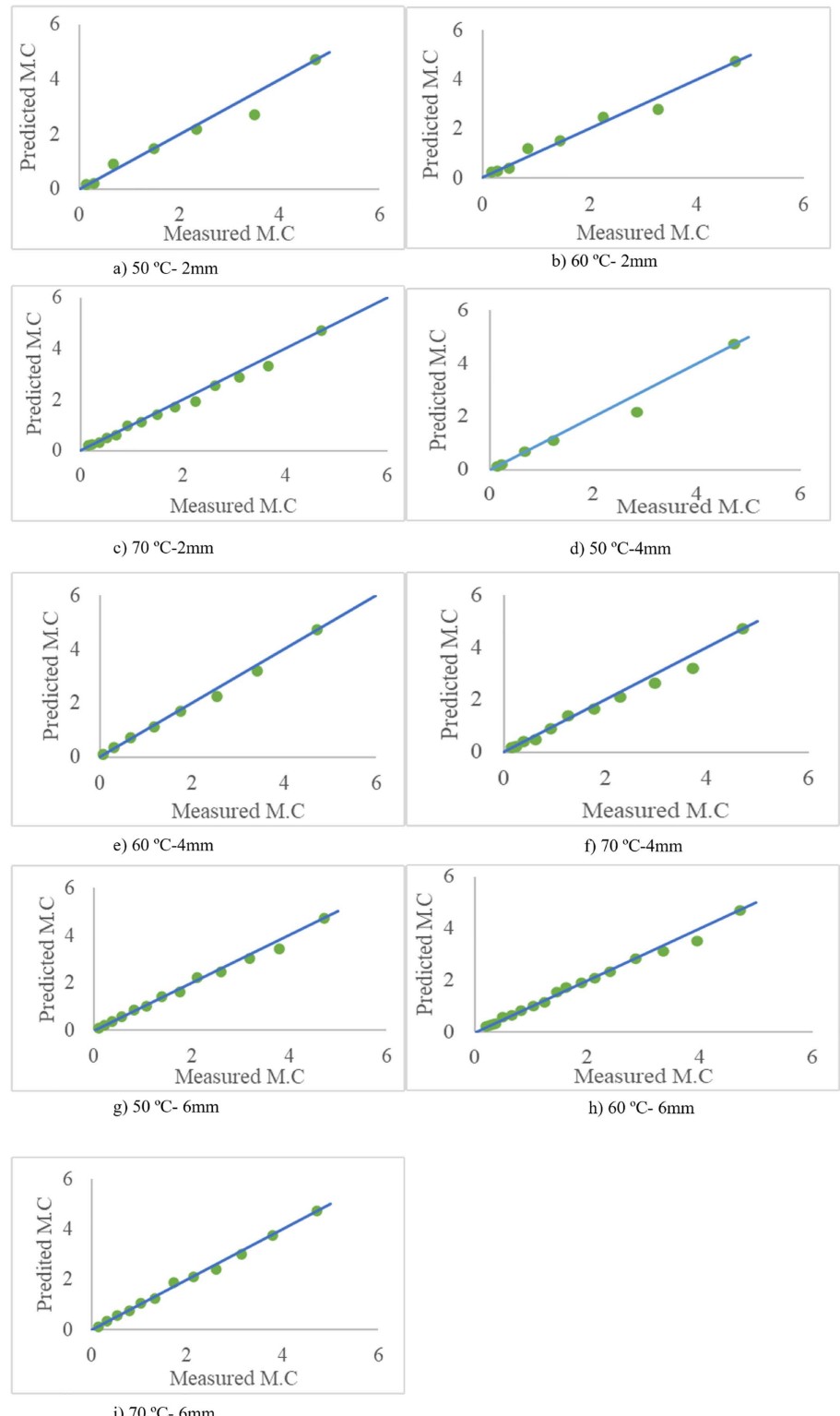

a) 50 ºC- 2mm

b) 60 ºC- 2mm

c) 70 ºC-2mm

d) 50 ºC-4mm

e) 60 ºC-4mm

f) 70 ºC-4mm

g) 50 ºC- 6mm

h) 60 ºC- 6mm

i) 70 ºC- 6mm

**Fig 14. Comparison Between Measured and Predicted Moisture Content for Different Experiments.**

**Table 6. Statistical comparison of experimental and modeled drying data.**

| Temperature | Slice thickness | $R^2$ | RMSE | MAE |
|---|---|---|---|---|
| 50 | 2 | 0.976 | 0.239 | 0.164 |
| 50 | 4 | 0.99 | 0.155 | 0.113 |
| 50 | 6 | 0.993 | 0.120 | 0.068 |
| 60 | 2 | 0.962 | 0.299 | 0.186 |
| 60 | 4 | 0.987 | 0.202 | 0.139 |
| 60 | 6 | 0.994 | 0.127 | 0.081 |
| 70 | 2 | 0.979 | 0.275 | 0.146 |
| 70 | 4 | 0.993 | 0.142 | 0.098 |
| 70 | 6 | 0.995 | 0.092 | 0.063 |

theoretical modeling of particulate solids based on diffusion modeling was conducted, demonstrating strong agreement between the experimental and predicted results [35].

## Conclusion

This study comprehensively examined the drying kinetics, moisture diffusivity, and shrinkage behavior of apple slices under varying drying conditions, specifically temperature and slice thickness. The results indicate that higher drying temperatures significantly reduce drying time and moisture content, while also minimizing shrinkage. Conversely, increasing slice thickness prolongs drying time due to greater resistance to moisture diffusion. The interplay between these factors reveals that higher temperatures mitigate the impact of slice thickness, leading to more efficient drying. Finite Element Modeling (FEM) effectively predicted moisture diffusion and structural changes during drying, offering a reliable tool for process optimization. The developed multivariate models demonstrated strong predictive capabilities for shrinkage and moisture diffusivity as functions of temperature and thickness, enabling improved control over drying processes. Furthermore, the color analysis indicated that drying at 70°C with a slice thickness of 4 mm resulted in minimal color changes, effectively balancing drying efficiency with product quality. These findings provide actionable insights for industrial applications, such as designing energy-efficient drying systems that reduce processing time, ensure uniform product quality, and minimize shrinkage-related losses. Future research could explore alternative drying methods and hybrid technologies to further enhance efficiency and product characteristics. Incorporating anisotropic shrinkage and variable tissue properties into FEM models, along with extending simulations to three dimensions, would enable more accurate predictions of moisture and temperature distributions, particularly in thicker slices. Additionally, SEM analysis can be employed to examine shrinkage behavior in dried materials for verification purposes. Experimental validation of such advanced models using non-destructive imaging or real-time moisture sensors could further improve their reliability and practical applicability in industrial drying processes.

## Supporting information

**S1 Table. Activation Energy.**
(XLSX)

**S2 Table. Color change.**
(XLSX)

**S3 Table. Drying Kinetics.**
(XLSX)

**S4 Table. Drying time.**
(XLSX)

**S5 Table. Effective diffusivity.**
(XLSX)

**S6 Table. Mean comarison of shrinkage.**
(XLSX)

**S7 Table. Modeling.**
(XLSX)

**S8 Table. Shrinkage Modeling.**
(XLSX)

**S1 Data. Activation Energy.**
(XLSX)

**S2 Data. Color change.**
(XLSX)

**S3 Data. Drying kinetics.**
(XLSX)

**S4 Data. Drying time.**
(XLSX)

**S5 Data. Effective diffusivity.**
(XLSX)

**S6 Data. Mean comarison of shrinkage.**
(XLSX)

**S7 Data. Modeling.**
(XLSX)

**S8 Data. Shrinkage modeling.**
(XLSX)

## Author contributions

**Conceptualization:** Mehdi Moradi.

**Data curation:** Sadegh Rashidi.

**Investigation:** Mehdi Moradi, Reza Raeesi, Sadegh Rashidi.

**Methodology:** Mehdi Moradi.

**Project administration:** Mehdi Moradi.

**Software:** Mehdi Moradi.

**Writing – original draft:** Mehdi Moradi.

**Writing – review & editing:** Mahdi Keramat-Jahromi.

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
