## [Decision Letter · Decision Letter 0]

3 Nov 2025

Dear Dr. Moradi,

Thank you for submitting your manuscript to PLOS ONE. After careful consideration, we feel that it has merit but does not fully meet PLOS ONE’s publication criteria as it currently stands. Therefore, we invite you to submit a revised version of the manuscript that addresses the points raised during the review process.

We look forward to receiving your revised manuscript.

Kind regards,

Andrey Nagdalian

Academic Editor

PLOS ONE

Journal Requirements:

3. In the online submission form, you indicated that [the data may be available after journal request.].

4. We note that Figures 1, 2, and 3 in your submission may contain copyrighted images. All PLOS content is published under the Creative Commons Attribution License (CC BY 4.0), which means that the manuscript, images, and Supporting Information files will be freely available online, and any third party is permitted to access, download, copy, distribute, and use these materials in any way, even commercially, with proper attribution. For more information, see our copyright guidelines: http://journals.plos.org/plosone/s/licenses-and-copyright.

1. You may seek permission from the original copyright holder of Figures 1, 2, and 3 to publish the content specifically under the CC BY 4.0 license.

Reviewers' comments:

Reviewer's Responses to Questions

**Comments to the Author**

1. Is the manuscript technically sound, and do the data support the conclusions?

Reviewer #1: Partly

Reviewer #2: Partly

2. Has the statistical analysis been performed appropriately and rigorously?

Reviewer #1: Yes

Reviewer #2: Yes

3. Have the authors made all data underlying the findings in their manuscript fully available?

Reviewer #1: No

Reviewer #2: Yes

4. Is the manuscript presented in an intelligible fashion and written in standard English?

Reviewer #1: Yes

Reviewer #2: Yes

Reviewer #1: The authors have undertaken a rigorous experimental and modeling approach to investigate the complex interplay between drying temperature, apple slice thickness, and the resulting kinetics, shrinkage, and color changes. However, the manuscript currently suffers from a perceived lack of innovation. While the study is technically sound, it does not sufficiently articulate its unique contribution to the field. Many studies have explored the effects of temperature and thickness on apple drying. The authors should work on more clearly showing the innovation of their work compared with existing studies in the literature. Is it the specific combination of models? Highlighting this novelty is critical. While the findings are valuable, their impact is diminished without a clear statement of what this work achieves that previous studies have not. The following detailed comments are provided to assist the authors in strengthening the paper, with a particular focus on better contextualizing its innovative aspects, to elevate it for publication.

Comment 1:

On page 11 (line 113), the text states the experiments were conducted in the "summer of 2024." Please correct this expression.

Comment 2:

On page 12 (line 146), the text refers to "Equation 5," but the corresponding equation is numbered as (4) (line 155). Please ensure all in-text citations to equations, figures, and tables are accurate.

Comment 3:

The Introduction mentions the use of the "Golden Delicious" apple variety but does not explain why this specific cultivar was chosen. Providing this context strengthens the experimental design. Please add a sentence in the Introduction or Materials and Methods section explaining the rationale for selecting Golden Delicious apples and what make this varietyan ideal model for this study.

Comment 4:

Please perform an Analysis of Variance (ANOVA) on the total color difference (ΔE) values presented in Figure 14. This will statistically validate whether the differences in color change across the various drying treatments are significant, adding a quantitative layer to your qualitative discussion.

Comment 5:

In the discussion of the multivariate regression model for moisture diffusivity (page 22, lines 339-344), the interpretation could be clearer and more direct. Please rephrase the analysis of the T² and T terms to be more intuitive.

Comment 6:

The explanation for why thinner slices have higher activation energy (page 23, lines 361-364) is slightly counter-intuitive and could be clarified. The current text states that thinner slices lose moisture quickly, meaning "more energy is needed for moisture to diffuse." Please refine this explanation and clarify the discussion on Activation Energy.

Comment 7:

The manuscript briefly mentions the assumption of isotropic shrinkage as a limitation (page 27). This section could be expanded to provide more specific directions for future research. In the final paragraph of the "Statistical Analysis" section or in the "Conclusion," please elaborate on this and suggest specific future work.

Comment 8:

Several figures and tables are difficult to read due to small font sizes, which will be a problem in the final publication. Please increase the font size for axis titles, labels, and legends in all graphs, especially Figures 15 and 16. Ensure that all text within tables is legible and that the table titles (e.g., Table 2) clearly describe the contents, including the statistical test used (e.g., "ANOVA results for the effect...").. Consider improving Readability!

Comment 9:

Please structure the abstract to clearly separate the key findings from their implications. For instance, explicitly state the main quantitative results and conclude with a clear statement on the optimal conditions found.

Comment 10:

Please carefully proofread the entire manuscript to correct minor typos and grammatical errors. For example, on page 9 (line 70), there is a stray "y" at the start of a sentence. A thorough check will ensure the final version is polished.

Reviewer #2: The manuscript titled “Predictive Modeling of Apple Slice Drying: Integrating Temperature, Thickness, and Shrinkage Dynamics” presents valuable experimental and modeling insights, but several technical clarifications and refinements are needed for strengthening the scientific quality. Kindly check attachment for comments.

**Do you want your identity to be public for this peer review?** For information about this choice, including consent withdrawal, please see our For information about this choice, including consent withdrawal, please see our Privacy Policy .

Reviewer #1: **Yes:** MERIEM ADNOUNIMERIEM ADNOUNI

Reviewer #2: No

---

## [Author Response · Author response to Decision Letter 1]

21 Nov 2025

Dear Editor and Reviewers,

We sincerely thank you for your careful assessment of our manuscript and for the valuable comments and suggestions provided. We have carefully revised the manuscript in accordance with all recommendations. Below, we provide a detailed, point-by-point response outlining the changes made and clarifying how each comment has been addressed.

We appreciate your time and consideration and believe that the revisions have substantially improved the quality and clarity of our work. Please find our detailed responses below.

Editor comment:

We note that Figures 1, 2, and 3 in your submission may contain copyrighted images. All PLOS content is published under the Creative Commons Attribution License (CC BY 4.0), which means that the manuscript, images, and Supporting Information files will be freely available online, and any third party is permitted to access, download, copy, distribute, and use these materials in any way, even commercially, with proper attribution.

Response: Thank you for the notification. We have replaced Figure 1 with an original schematic illustration and have removed Figures 2 and 3 to ensure that all figures comply with CC BY 4.0 licensing requirements.

Reviewer 1:

Predictive Modeling of Apple Slice Drying: Integrating Temperature, Thickness, and Shrinkage Dynamics

The authors have undertaken a rigorous experimental and modeling approach to investigate the complex interplay between drying temperature, apple slice thickness, and the resulting kinetics, shrinkage, and color changes. However, the manuscript currently suffers from a perceived lack of innovation. While the study is technically sound, it does not sufficiently articulate its unique contribution to the field. Many studies have explored the effects of temperature and thickness on apple drying. The authors should work on more clearly showing the innovation of their work compared with existing studies in the literature. Is it the specific combination of models? Highlighting this novelty is critical. While the findings are valuable, their impact is diminished without a clear statement of what this work achieves that previous studies have not. The following detailed comments are provided to assist the authors in strengthening the paper, with a particular focus on better contextualizing its innovative aspects, to elevate it for publication.

Response: Thank you for your suggestion. The Introduction has been revised to clearly highlight the novelty of this work. The revised text emphasizes that this study uniquely integrates experimental measurements with predictive modeling, combining multivariate regression and Finite Element Modeling (FEM) to simultaneously quantify drying kinetics, shrinkage, moisture diffusivity, and color changes. Predictive models were developed to express shrinkage and moisture diffusivity explicitly as functions of slice thickness and temperature, providing a reliable, quantitative tool for optimizing drying parameters. This integrated approach enables accurate prediction of both physical and quality-related changes under varying conditions, offering actionable guidance for designing energy-efficient drying systems—a contribution that has not been addressed in previous studies.

Comment 1:

On page 11 (line 113), the text states the experiments were conducted in the "summer of 2024." Please correct this expression.

Response: Thank you for your comment. The experiments were indeed performed in June 2024, before the manuscript was submitted. The sentence has been revised to: ‘After setting up the system, the drying experiments were carried out in June 2024 at the Department of Biosystems Mechanical Engineering, Shiraz University.

Comment 2:

On page 12 (line 146), the text refers to "Equation 5," but the corresponding equation is numbered as (4) (line 155). Please ensure all in-text citations to equations, figures, and tables are accurate.

Response: Thank you for your comment. All equation, figure, and table numbers have been checked and updated as appropriate.

Comment 3:

The Introduction mentions the use of the "Golden Delicious" apple variety but does not explain why this specific cultivar was chosen. Providing this context strengthens the experimental design. Please add a sentence in the Introduction or Materials and Methods section explaining the rationale for selecting Golden Delicious apples and what make this varietyan ideal model for this study.

Response: Thank you for your comment. A sentence has been added to the Materials and Methods explaining the rationale for selecting the ‘Golden Delicious’ cultivar.

“The Golden Delicious apple was chosen due to its global commercial relevance, uniform morphology, and stable chemical composition, which minimize variability and make it a suitable model cultivar for evaluating drying performance and quality changes.”

Comment 4:

Please perform an Analysis of Variance (ANOVA) on the total color difference (ΔE) values presented in Figure 14. This will statistically validate whether the differences in color change across the various drying treatments are significant, adding a quantitative layer to your qualitative discussion.

Response: Thank you for your suggestion. An ANOVA was conducted to evaluate the effects of drying temperature, slice thickness, and their interaction on the total color difference (ΔE). The analysis revealed that temperature, thickness, and their interaction all significantly affected ΔE (p < 0.0001). These results have been included in Table 4 and discussed in the revised manuscript to provide a quantitative assessment of color change across the drying treatments.

Comment 5:

In the discussion of the multivariate regression model for moisture diffusivity (page 22, lines 339-344), the interpretation could be clearer and more direct. Please rephrase the analysis of the T² and T terms to be more intuitive.

Response: Thank you for this suggestion. The paragraph discussing the regression model has been revised to provide a clearer and more intuitive interpretation of the temperature terms (T² and T). The updated text explains that these terms together describe a non-linear effect of temperature on moisture diffusivity, with Deff generally increasing at higher temperatures. Slice thickness effects have also been clarified, emphasizing that thicker slices lead to higher Deff. The revised paragraph improves readability while maintaining the technical accuracy of the model.

Comment 6:

The explanation for why thinner slices have higher activation energy (page 23, lines 361-364) is slightly counter-intuitive and could be clarified. The current text states that thinner slices lose moisture quickly, meaning "more energy is needed for moisture to diffuse." Please refine this explanation and clarify the discussion on Activation Energy.

Response: Thank you for this valuable comment. The paragraph discussing activation energy has been revised to provide a clearer and more intuitive explanation. The revised text clarifies that thinner slices have a larger surface-to-volume ratio, which leads to faster initial moisture loss at the surface. As a result, maintaining effective moisture transport in thinner slices requires more energy, which is reflected in their higher activation energy values. In contrast, thicker slices dry more slowly and uniformly, making their moisture diffusion less sensitive to temperature changes. The revised paragraph also highlights supporting findings from previous studies to contextualize these observations.

Comment 7:

The manuscript briefly mentions the assumption of isotropic shrinkage as a limitation (page 27). This section could be expanded to provide more specific directions for future research. In the final paragraph of the "Statistical Analysis" section or in the "Conclusion," please elaborate on this and suggest specific future work.

Response: Thank you for your suggestion. The Conclusion has been revised to provide a more detailed discussion of the limitations related to isotropic shrinkage and directions for future research. The updated text now highlights the potential for incorporating anisotropic shrinkage effects and variable tissue properties, extending the model to three dimensions, and performing experimental validation using non-destructive imaging or real-time moisture sensors. These additions provide specific and actionable directions for future studies and enhance the clarity and applicability of the conclusions.

Comment 8:

Several figures and tables are difficult to read due to small font sizes, which will be a problem in the final publication. Please increase the font size for axis titles, labels, and legends in all graphs, especially Figures 15 and 16. Ensure that all text within tables is legible and that the table titles (e.g., Table 2) clearly describe the contents, including the statistical test used (e.g., "ANOVA results for the effect...").. Consider improving Readability!

Response: Thank you. The font size of the figures has been increased, and the titles of the related tables have been revised.

Comment 9:

Please structure the abstract to clearly separate the key findings from their implications. For instance, explicitly state the main quantitative results and conclude with a clear statement on the optimal conditions found.

Response: Thank you for the suggestion. The abstract has been revised to clearly highlight the key findings and their implications in a single, coherent paragraph. Quantitative results such as the effects of temperature and slice thickness on drying time, shrinkage, and moisture diffusivity are explicitly stated. The optimal drying conditions (70°C and 4 mm slice thickness) are also clearly indicated, along with recommendations for future research, including anisotropic shrinkage modeling, energy analysis, and hybrid drying techniques. This revision improves clarity and ensures that both the main results and their significance are immediately understandable to the reader.

Comment 10:

Please carefully proofread the entire manuscript to correct minor typos and grammatical errors. For example, on page 9 (line 70), there is a stray "y" at the start of a sentence. A thorough check will ensure the final version is polished.

Response:

Thank you for pointing this out. The entire manuscript has been thoroughly proofread, and all identified typographical and grammatical errors, including the stray “y” on page 9, line 70, have been corrected. Additional revisions were made to improve clarity, readability, and consistency throughout the text.

Reviewer 2

Comment 1

The abstract is informative, but it would benefit from explicitly mentioning the experimental replication and statistical approach used for validation.

Response:

Thank you for the helpful suggestion. The abstract has been revised to include a clear statement that all drying experiments were performed in triplicate and that statistical validation, including Analysis of Variance (ANOVA), was used to assess the significance of the effects of temperature and slice thickness. This addition strengthens the methodological clarity of the abstract and highlights the rigor of the study’s experimental and analytical approach.

Commnet2:

The conclusion on “optimal visual quality at higher temperatures” seems counterintuitive; please justify with supporting data or revise the phrasing.

Response: Thank you for this observation. We agree that stating “optimal visual quality at higher temperatures” in the abstract may appear counterintuitive without context. We have revised the abstract to clarify that the improved visual quality was specifically associated with the combination of 70°C and a 4 mm slice thickness, which resulted in the lowest color change (ΔE) among all treatments. This combination reduced overall drying time enough to limit enzymatic browning, despite the high temperature. The revised phrasing now clearly reflects this interaction and avoids any misleading generalization about high-temperature drying.

Comment3: The introduction provides background, but it should briefly highlight gaps in existing FEM-based drying studies for apples to justify novelty.

Response: Thanks for your comment. In this regard, we developed the last section of introduction to highlight the novelty of the research, in use of FEM to simultaneously quantify multiple aspects of the drying process. The revised section is as following:

“This study investigates the effects of slice thickness and drying temperature on the drying kinetics, moisture diffusivity, shrinkage, and color of apple slices. A key innovation of this work lies in the integration of experimental measurements with predictive modeling, combining multivariate regression and Finite Element Modeling (FEM) to simultaneously quantify multiple aspects of the drying process. Predictive models were developed to express shrinkage and moisture diffusivity explicitly as functions of slice thickness and temperature, providing a reliable, quantitative tool for optimizing drying parameters and improving both efficiency and product quality. Unlike previous studies, which often focus on individual factors or qualitative trends, this integrated approach enables accurate prediction of physical and quality-related changes under varying operational conditions, offering actionable insights for designing energy-efficient drying systems tailored to specific product characteristics.”

Comment4: The statement that thickness has a greater effect than temperature needs citations of at least two prior studies to reinforce the claim.

Response: Thank you for this helpful comment. We have now added supporting literature that directly demonstrates the comparatively stronger influence of slice thickness on drying time. Studies on melon and G. erubescens fruits show that increasing slice thickness results in a substantially greater increase in drying duration than the reductions achieved by increasing temperature. In this regard, following paragraph was added: “Evidence from previous drying studies supports the greater influence of slice thickness compared to temperature on drying time. For instance, in the drying of melon slices, increasing thickness from 3 mm to 5 mm led to increases of 38–87% in drying time across temperatures of 60, 70, 80, and 90°C, whereas raising the air temperature within a constant thickness reduced drying time by only 40–53% [18]. Similarly, in studies on other fruit matrices such as G. erubescens, thicker slices consistently required substantially longer drying periods due to the increased moisture diffusion path [19]. These findings collectively demonstrate that slice thickness exerts a stronger limiting effect on moisture removal than temperature, reinforcing our observation that thickness has a more pronounced impact on drying time.”

Comment 5

The detailed hardware description (brands, model numbers) is good, but airflow distribution uniformity inside the cabinet should be validated, as it significantly affects results.

Response: We appreciate this valuable observation. In response, we have added a clarification in the Materials and Methods section describing how airflow uniformity was checked prior to experiments. Specifically, airflow velocity was measured at multiple points across the tray area using an anemometer to confirm spatial uniformity. These measurements confirmed that the airflow variation across the drying chamber remained within ±5%, indicating sufficient uniform airflow for reliable drying experiments. This information has been incorporated into the revised manuscript to address the reviewer’s concern.

Comment6

Replication strategy (triplicate runs) is mentioned; however, error bars or confidence intervals should be presented in figures to reflect variability.

Response: Thank you. Error bars have been added to the figures.

Comment7

The assumption of constant diffusivity in FEM needs a stronger justification; real food tissues often show variable diffusivity during drying.

Response:

Thank you for this valuable comment. We would like to clarify that the FEM simulations in this study did not assume constant moisture diffusivity. The discussion of constant diffusivity in the theoretical modeling section was included only to explain why analytical solutions tr

---

## [Decision Letter · Decision Letter 1]

21 Dec 2025

Dear Dr. Mehdi Moradi,

Thank you for submitting your manuscript to PLOS ONE. After careful consideration, we feel that it has merit but does not fully meet PLOS ONE’s publication criteria as it currently stands. Therefore, we invite you to submit a revised version of the manuscript that addresses the points raised during the review process.

We look forward to receiving your revised manuscript.

Kind regards,

Andrey Nagdalian

Academic Editor

PLOS One

Journal Requirements:

Additional Editor Comments:

Dear authors, I am pleased to inform you that both reviewers now recommend publication of the revised manuscript. However, the Reviewer 1 left some additional minor comments that will help to further improve the text of the manuscript. Please consider them and make revision thoroughly.

Reviewers' comments:

Reviewer's Responses to Questions

**Comments to the Author**

Reviewer #1: All comments have been addressed

Reviewer #2: All comments have been addressed

2. Is the manuscript technically sound, and do the data support the conclusions?

Reviewer #1: Yes

Reviewer #2: Yes

3. Has the statistical analysis been performed appropriately and rigorously?

Reviewer #1: Yes

Reviewer #2: Yes

4. Have the authors made all data underlying the findings in their manuscript fully available?

Reviewer #1: Yes

Reviewer #2: Yes

5. Is the manuscript presented in an intelligible fashion and written in standard English?

Reviewer #1: Yes

Reviewer #2: Yes

Reviewer #1: For a journal like PLOS ONE, which does not perform copy-editing, clarity and correctness in English are essential. By addressing the following points, the authors can significantly improve the quality of the English prose, making the manuscript more professional, easier to read, and more suitable for publication:

1. The phrase "(is for slice thickness of 2 mm)" is grammatically incorrect and awkwardly inserted. It breaks the flow of the sentence. Rephrase for clarity. For example: "As shown in Figure 15 for a slice thickness of 2 mm, the solution stabilized..." or "Figure 15 shows the results for a 2 mm slice thickness, where the solution stabilized...".

2. Throughout the manuscript: There are minor inconsistencies, such as the use of "et al" versus "et al." (the latter, with a period, is standard). Also, check for consistent capitalization in titles and headings.

3. "The theoretical model exhibited a high correlation coefficient (R²) ranging from 0.99 to 0.97, with a MAE between 0.063 and 0.186, and a RMSE ranging from 0.092 to 0.275." The sentence is long and lists many statistics. While grammatically correct, it could be clearer.

Correction (for better readability): "The theoretical model demonstrated a high correlation with experimental data, achieving R² values between 0.97 and 0.99. The model's accuracy was further confirmed by low Mean Absolute Error (MAE) values (0.063–0.186) and Root Mean Square Error (RMSE) values (0.092–0.275)."

Reviewer #2: The manuscript has been thoroughly and carefully revised according to the comments and is now suitable for acceptance in its current form.

**Do you want your identity to be public for this peer review?** For information about this choice, including consent withdrawal, please see our For information about this choice, including consent withdrawal, please see our Privacy Policy .

Reviewer #1: **Yes:** MERIEM ADNOUNIMERIEM ADNOUNI

Reviewer #2: No

---

## [Author Response · Author response to Decision Letter 2]

30 Dec 2025

Response to Reviewers

We sincerely thank the reviewers for their careful evaluation of our manuscript and for their constructive and insightful comments. We have revised the manuscript thoroughly to improve clarity, consistency, and overall quality. All changes have been incorporated into the revised version. Our detailed responses are provided below.

Reviewer #1:

For a journal like PLOS ONE, which does not perform copy-editing, clarity and correctness in English are essential. By addressing the following points, the authors can significantly improve the quality of the English prose, making the manuscript more professional, easier to read, and more suitable for publication:

Response: We appreciate the reviewer’s emphasis on clarity and correctness in English, which is particularly important for PLOS ONE. The manuscript has been carefully revised to address all the points raised.

Comment 1. The phrase "(is for slice thickness of 2 mm)" is grammatically incorrect and awkwardly inserted. It breaks the flow of the sentence. Rephrase for clarity. For example: "As shown in Figure 15 for a slice thickness of 2 mm, the solution stabilized..." or "Figure 15 shows the results for a 2 mm slice thickness, where the solution stabilized...".

Response: Thanks. We agree with the reviewer and have revised the sentence to improve grammatical correctness and readability. The phrase has been reworded and smoothly integrated into the sentence.

Revision made:

The sentence now reads:

“As shown in Figure 13 for a slice thickness of 2 mm, the solution stabilized..." or "Figure 15 shows the results for a 2 mm slice thickness, where the solution stabilized...”

Comment 2. Throughout the manuscript: There are minor inconsistencies, such as the use of "et al" versus "et al." (the latter, with a period, is standard). Also, check for consistent capitalization in titles and headings.

Response:

Thank you for pointing this out. We have carefully reviewed the entire manuscript and corrected all instances to ensure consistency.

Comment 3. "The theoretical model exhibited a high correlation coefficient (R²) ranging from 0.99 to 0.97, with a MAE between 0.063 and 0.186, and a RMSE ranging from 0.092 to 0.275." The sentence is long and lists many statistics. While grammatically correct, it could be clearer.

Correction (for better readability): "The theoretical model demonstrated a high correlation with experimental data, achieving R² values between 0.97 and 0.99. The model's accuracy was further confirmed by low Mean Absolute Error (MAE) values (0.063–0.186) and Root Mean Square Error (RMSE) values (0.092–0.275)."

Response:

We agree with the reviewer’s suggestion and have adopted the revised wording to improve clarity and readability.

Revision made:

The sentence has been revised to:

“The theoretical model demonstrated a high correlation with experimental data, achieving R² values between 0.97 and 0.99. The model’s accuracy was further confirmed by low Mean Absolute Error (MAE) values (0.063–0.186) and Root Mean Square Error (RMSE) values (0.092–0.275).”

Reviewer #2: The manuscript has been thoroughly and carefully revised according to the comments and is now suitable for acceptance in its current form.

Response:

We sincerely thank the reviewer for their positive evaluation and support. We appreciate the time and effort invested in reviewing our work.

---

## [Editor Report · Decision Letter 2]

7 Jan 2026

Predictive Modeling of Apple Slice Drying: Integrating Temperature, Thickness, and Shrinkage Dynamics

PONE-D-25-36704R2

Dear Dr. Moradi,

We’re pleased to inform you that your manuscript has been judged scientifically suitable for publication and will be formally accepted for publication once it meets all outstanding technical requirements.

Kind regards,

Andrey Nagdalian

Academic Editor

PLOS ONE

---

## [Editor Report · Acceptance letter]

PONE-D-25-36704R2

PLOS One

Dear Dr. Moradi,

I'm pleased to inform you that your manuscript has been deemed suitable for publication in PLOS One. Congratulations! Your manuscript is now being handed over to our production team.

Kind regards,

on behalf of

Dr. Andrey Nagdalian

Academic Editor

PLOS One